# Synthesis, Cyclooxygenases Inhibition Activities and Interactions with BSA of *N*-substituted 1*H*-pyrrolo[3,4-c]pyridine-1,3(*2H*)-diones Derivatives

**DOI:** 10.3390/molecules25122934

**Published:** 2020-06-25

**Authors:** Edward Krzyżak, Dominika Szkatuła, Benita Wiatrak, Tomasz Gębarowski, Aleksandra Marciniak

**Affiliations:** 1Department of Inorganic Chemistry, Wroclaw Medical University, ul. Borowska 211a, 50–556 Wrocław, Poland; aleksandra.marciniak@umed.wroc.pl; 2Department of Medicinal Chemistry, Wroclaw Medical University, Borowska 211, 50–556 Wroclaw, Poland; dominika.szkatula@umed.wroc.pl; 3Department of Basic Medical Sciences, Wroclaw Medical University, Borowska 211, 50–556 Wroclaw, Poland; benita.wiatrak@umed.wroc.pl (B.W.); tomasz.gebarowski@umed.wroc.pl (T.G.)

**Keywords:** pyrrolo-pyridine derivatives, cyclooxygenase, anti-inflammatory, serum albumin interactions, fluorescence quenching, molecular docking

## Abstract

Inhibition of cyclooxygenase is the way of therapeutic activities for anti-inflammatory pharmaceuticals. Serum albumins are the major soluble protein able to bind and transport a variety of exogenous and endogenous ligands, including hydrophobic pharmaceuticals. In this study, a novel *N*-substituted 1*H*-pyrrolo[3–c]pyridine-1,3(2*H*)-diones derivatives were synthesized and biologically evaluated for their inhibitory activity against cyclooxygenases and interactions with BSA. In vitro, COX-1 and COX-2 inhibition assays were performed. Interaction with BSA was studied by fluorescence spectroscopy and circular dichroism measurement. The molecular docking study was conducted to understand the binding interaction of compounds in the active site of cyclooxygenases and BSA. The result of the COX-1 and COX-2 inhibitory studies revealed that all the compounds potentially inhibited COX-1 and COX-2. The IC_50_ value was found similar to meloxicam. The intrinsic fluorescence of BSA was quenched by tested compounds due to the formation of A/E–BSA complex. The results of the experiment and molecular docking confirmed the main interaction forces between studied compounds and BSA were hydrogen bonding and van der Waals force.

## 1. Introduction

Derivatives of 3,4-pyridinedicarboximides have been interesting for many years. Most of them show various kinds of biological activities. For example, *N*-(2,6-dimethylphenyl)-3,4- pyridinedicarboximide is active against MES (Maximal Electroshock Seizures) and appears to be a promising compound for the design of anticonvulsant drugs [1]. Some of [pyrrolo[3,4-c]pyridin-1,3(2*H*)-dion-2-yl] acetic acids derivatives show significant AR (aldose reductase) inhibitory activity [2]. The 3,4-pyridinedicarboximide was a base compound in the design and synthesis of azaisoindolinone derivatives with a lipophilic chain. These compounds were designed as InhA inhibitors and as anti-mycobacterium tuberculosis agents [3]. The derivatives of bicyclic hydroxy-1*H*-pyrrolopyridine-triones were designed as a new family of HIV-1 integrase inhibitors [4].

Cyclooxygenase (COX) is the key enzyme required for the conversion of arachidonic acid to prostaglandins. At least two isoforms of COX are known—COX-1 and COX-2. COX-1 is involved in the synthesis of prostaglandins responsible for maintaining normal body function in the kidney, GIT, and other organs. COX-2 is the isoform that plays a major part in the inflammatory process and the pain associated with it [5,6]. COX-1 and COX-2 are the therapeutic targets for drugs, including ibuprofen, naproxen, diclofenac, or piroxicam, as well as newer COX-2 selective inhibitors. Through their anti-inflammatory, anti-pyretic, and analgesic activities, they represent a choice treatment in various inflammatory diseases such as arthritis, rheumatisms as well as relieving the pains of everyday life [7,8].

From a biopharmaceutical point of view, one of the most important biological functions of albumins is their ability to carry drugs. The drug-BSA (bovine serum albumin) or HSA (human serum albumin) interaction may result in the formation of a stable complex, which has a great impact on discovering pharmacokinetic and pharmacodynamics implications. Spectroscopic methods like fluorescence, UV-vis, and circular dichroism spectroscopies help acquire this knowledge.

A 2-[2-hydroxy-3-(4-aryl-1-piperazinyl)propyl] derivatives of 4-methoxy- and 4-ethoxy-6-methyl-1*H*-pyrrolo[3,4-c]pyridine-1,3(2*H*)-diones indicate strong analgesic properties (stronger than ASA and Morphine in writhing syndrome test), and they are non-toxic [9]. The compounds **A**–**C** presented in this work were designed by some modifications: shortening the chain between imide nitrogen and nitrogen of piperazine ring, elimination of the carboxyl group, and movement of the methyl group to 5 position (Figure 1). The compounds **D** and **E** are smaller and do not have a phenylo-piperazino ring. The compound A was previously synthesized and tested for anxiolytic effects [10]. However, no activity was found, only the potential analgesic effect was suggested. The aim of this work was the investigation of the synthesized compounds for their potencies to inhibit COX-1 and COX-2 enzymes and study the interaction with bovine serum albumin.

## 2. Results and Discussion

### 2.1. Chemistry

Synthetic scheme of the studied compounds, as shown in Scheme 1. The compounds **2**, **3** and **A** were obtained according to the literature data [10,11,12,13]. Imide **2** is the starting product for derivatives **A**, **B** and **C** and many other substances described earlier [10,14,15]. Compounds **2** and **3** differ slightly in structure, imide 3 contains a methoxy substituent attached to the carbon atom in a position adjacent to the nitrogen atom of the pyridine ring, while derivative **2** is its *N*-methylanalogue. To compare the biological properties of both basic systems, it was decided to synthesize analogues of the previously described derivatives containing the same amino residues: 1-phenylpiperazine (**A**), 1-(2-methoxyphenyl)piperazine (**B**) and 1-(3-trifluoromethyphenyl)piperazine (**C**) [9]. The presence of an acidic proton atom at the imide nitrogen atom in position 2 makes it possible to carry out the aminomethylation reaction according to the mechanism described by Mannich [13,16]. The reaction was carried out using aqueous solution of formaldehyde (HCHO) and N-arylpiperazines (commercial products Sigma-Aldrich) at the reflux temperature of tetrahydrofuran (THF) for several hours (method I, Scheme 1). The course of condensation was monitored by TLC. During the reaction of (**A** and **B**) or after the evaporation of the solvent (**C**) final products were obtained. Poor solubility in organic solvents (chloroform, ethanol, methanol, ethyl acetate) and their mixtures prevented the purification of the products by column chromatography, therefore the final compounds were tested in the form of crude products. NMR spectral analysis was performed in dimethyl sulfoxide (DMSO). All reactions proceeded with a very good yield of 79.4–93.8%.

Imid **3**, previously described [11,13] was the starting product for the synthesis of several dozen derivatives with confirmed biological activity (Śladowska et al.) [9,13,17,18].

The main line of analogues was based on the structure of 4-alkoxy-1*H*-pyrrolo(3,4-c) pyridine-1,3(2*H*) -dione. Optimal biological properties in the behavioral tests showed imide**1**: 4-alkoxy-*N*- [3- (*N*-phenyl-4-piperazinyl) -2-hydroxy] propyl-1*H*-pyrrolo (3,4-c) pyridine-1, 3(2*H*)-dione. One of the structure modifications is the assumption of shortening to two (**E**) or one (**D**) methylene groups of the alkyl linker between the 2-methoxy-3,4-pyridinedicarbox imide moiety and the arylamine residue. In order to obtain **D** and **E** derivatives, commercial preparations of 4-(2-chloroethyl)-morpholine and pyrrolidine were used, condensing the potassium salt of imide **3**, under the influence of potassium ethoxylate in anhydrous ethanol (method II, Scheme 1; **E**), or in the Mannich reaction-THF-solvent and formalin (method III, Scheme 1; **D**). The reaction yield was 50% (**E**) and 60% (**D**).

### 2.2. Viability of Cell Cultures

MTT assay showed the concentration-dependence of cytotoxicity of tested compounds, and cell viability decreased with increasing concentration (Figure 2). After incubation with compounds **A**–**D** at 10 μM and **A** and **D** at 50 μM, a statistically significant increase in proliferation was noted. In the presence of **D** and **E**, no cytotoxic activity was observed in the whole concentration range tested. For all compounds at each concentration, the decrease in culture viability was less than 30% compared to the control (i.e., no cytotoxic potential of the tested compounds).

### 2.3. Cyclooxygenase Inhibition

All compounds tested inhibited both COX-1 and COX-2 activity (Table 1). In the case of compounds **A** and **D**, COX-1 inhibition was stronger compared to meloxicam, which was the reference compound. Compounds with COX-2 inhibitory activity (except for **C**) showed higher activity than meloxicam. Most compounds tested (except for **D**) showed stronger COX-2 selectivity than meloxicam. The IC_50_ values obtained for meloxicam are significantly higher than those published by Ogino et al. [19]. However, it was an in vivo study in rats, and we performed only the enzymatic assay. IC_50_ values for tested compounds are similar to those of meloxicam. Therefore in the next stage of research, it is planned to conduct in vivo tests to check the anti-inflammatory properties and gastrointestinal toxicity of the most promising compound **E**.

### 2.4. Molecular Docking Studies

The empirical scoring function of iGEMDOCK is estimated as E_total_ = Van der Waal’s + hydrogen bonding + electrostatic interactions. The docking results are presented in Table 2. The results indicated that compound **D** showed the lowest total binding energy E_total_ with respect to COX-1. The highest value was observed for compound **E**. The docking scores showed a good correlation with IC_50_ values (Table 1). For interactions with COX-2 E_total_ was more negative. The lowest energy was observed for **A** and **E** and the highest for **C**. This result was also in good correlation with inhibitory activity. The compounds with the lowest binding energy showed the most promising biological activity in cyclooxygenase inhibition assay. The molecular docking results also indicated that binding energy of tested compounds were more negative than for meloxicam.

All of the synthesized compounds form 1 to 3 hydrogen bonds with Ser530, Arg120, Leu531, Tyr355 amino acid residues inside COX-1, COX-2 active site, and several hydrophobic interactions (Table 3, Figure 3**).** The COX ligand binding site has four characteristic subdomains **A**–**D** [20]. Subdomain **A** represents the mode of binding of flurbiprofen; subdomain **B** represents the mode of binding of meloxicam and piroxicam; subdomain **C** represents an entrance region of the enzyme binding domain, and subdomain **D** represents the position of the residue in position 523. The most active compound to inhibit COX-1, 4-Methoxy–*N*-[1-(*N*-pyrrolidine)-methyl]-6-methyl-1*H*-pyrrolo[3.4-c]pyridine-1,3(2*H*)-dione (**D**) is bound to enzyme by hydrogen bonds Leu 531 and several hydrophobic interactions, π-sigma with Ala527, π-alkyl with Val349, Leu352, Leu531, Ala527 (Table 3, Figure 3). The main part of the compound, with the pyrrolo-pyridine ring, there is in the subdomain **B** position, the pyrrolidine ring in the entrance region **A**. The position of the molecule is very similar to meloxicam (Figure 4). The most active compound with phenylo-piperazino ring 5,6-dimethyl-4-oxo-2-[(4-phenyl-1-piperazinyl)methyl]-1*H*-pyrrolo[3.4-c]pyridine-1,3(2*H*)-dione is also situated in subdomain **B** (Figure 4**).** It binds by *H*-binding with Ser350 and several hydrophobic interactions (Table 3). The most active compound to inhibit COX-2, 4-Methoxy–*N*-[2-(*N*-morpholine)-ethyl]-6-methyl-1*H*-[pyrrolo[3.4-c]pyridine-1,3(2*H*)-dione is bound to enzyme by hydrogen bonds with Arg120 and Tyr355, and it is stabilized by some several hydrophobic interactions (Figure 3, Table 3). The pyrrolo-pyridine ring takes the position in subdomain **B**, and the morpholine ring is closer region **A** (Figure 4). The most active compound with phenylo-piperazino ring 5,6-dimethyl-4-oxo-2-[(4-phenyl-1-piperazinyl)methyl]-1*H*-pyrrolo[3.4-c]pyridine-1,3(2*H*)-dione, also situated in the subdomain **B** (Figure 4). The orientation in the active site is very similar to the position in COX-1 pocket and meloxicam location.

### 2.5. Fluorescence Quenching of BSA by Compound ***A***–***E***

The steady-state fluorescence spectroscopy and the synchronous fluorescence spectroscopy were used to study fluorescence quenching of BSA by compounds **A**–**E**. The fluorescent behavior of BSA is due to the amino acid residues: Trp, Tyr, and Phe. However, Trp residue has the strongest fluorescence intensity. Thus, the two Trp residues of BSA are mainly responsible for its fluorescence. The fluorescence spectra were recorded for BSA in the presence of studied compounds at the excitation wavelengths λ = 280 nm (both Trp and Tyr residues are excited) and concentration range 0.0–5.0 µM. The fluorescence emission spectra for all tested compounds were shown in Figure 5. The fluorescence intensity of BSA was decreased with increasing concentration of compounds **A**–**E**. The presence of **A**–**E** not only quenched the fluorescence of BSA, but it also caused a blue shift in the maximum emission wavelength of protein. For that, a strong fluorescence emission band at 349 nm, was observed. A careful analysis of the spectra discloses that at a lower concentration of **A**–**E** (**A**–**E**/BSA molar ratio 0.2–2:1), its shift is rather slight, up to 1.5 nm. In the medium range of concentration, change in λ_max_ is more evident. **A** large change in λ_max_ in the presence of a high concentration of **A**–**E** (**A**–**E**/BSA molar ratio 5:1) indicates the unfolding of the polypeptide backbone of macromolecule [21]. The blue shift effect expressed that the conformation of BSA was changed. It also indicated the amino acid residues are located in a more hydrophobic environment and are less exposed to the solvent [22]. Fluorescence quenching and shift of λ_max_ identifies interaction with BSA and can suggest the formation of complexes (static quenching). However, it can also be the result of the collisional encounters (dynamic quenching). In order to confirm the quenching mechanism and complex formation, the fluorescence data were further analyzed by the Stern–Volmer equation and dependence on temperature.

Fluorescence intensities were corrected for the absorption of excitation light and re-absorption of emitted light to decrease the inner filter using the following relationship:(1)$$F_{corr}=F_{obs}10^{\frac{\left(A_{ex}+A_{em}\right)}{2}}$$
where, *F**_corr_* and *F**_obs_* are the corrected and observed fluorescence intensities, respectively. *A_ex_* and *A_em_* are the absorbance values at excitation and emission wavelengths, respectively.

In most cases, the possible quenching mechanism is characterized by a linear Stern–Volmer plot and is usually analyzed using the classical Stern–Volmer Equation (1) [23]:(2)$$\frac{F_{0}}{F}=1+k_{q}\tau_{0}\left[Q\right]=1+K_{sv}\left[Q\right]$$
where *F*_0_ and *F* are the steady-state fluorescence intensities at the maximum wavelength in the absence and presence of quencher, respectively, *k_q_* the quenching rate constant of the biomolecule, *τ*_0_ the average lifetime of the biomolecule, [*Q*] is the quencher concentration, and *K_sv_* is the Stern–Volmer constant. For determining the type of quenching liner fitting was analyzed (Figure 6). The average lifetime of the fluorophore in the excited state for a biomolecule is 10^−8^ s [24]. According to Equation (2), the Stern–Volmer constant and the quenching rate constants were obtained from the linear fitting of the experimental data. Received data are collected in Table 4. For dynamic quenching, the maximum scatters collision quenching constant of different quenchers with the biopolymers was reported to 2 × 10^10^ dm^3^·mol^−1^·s^−1^ [25]. The results showed that the value of k_q_ for all cases is much greater, which indicated that the probable quenching mechanism of fluorescence of BSA by **A**–**E** is not caused by a dynamic collision but from the formation of a complex.

The dynamic and static quenching can be distinguished by their dependence on temperature. The higher temperature may result in decreasing stability of the complex and thus smaller values of the static quenching constant. The fluorescence data were analyzed at three different temperatures, and the fluorescence quenching constant of BSA was calculated using the Stern–Volmer Equation (2). The results are listed in Table 4. The Stern–Volmer quenching constant K_sv_ is inversely correlated with temperature, and k_q_ is much greater than the value of the maximum scatter collision quenching constant. It suggested the formation of ground-state complex and involvement of static quenching between BSA and studied compounds.

### 2.6. Circular Dichroism Spectra

CD spectroscopy is a very good method to determine the conformational changes in the secondary structure of proteins in case of presence of compounds which can interact with protein molecule [26]. In this study, there were analyzed changes in the secondary structure of BSA in the absence and presence of **A**–**E** compounds. In all CD spectra, two negative bands were observed at near 208 nm and 222 nm, what is characteristic for BSA (Figure 7). It is a typical feature of the α-helical structure of the protein. Any change in this region of spectra suggests conformational changes in protein molecules [27]. Figure 7 shows that in the presence of all analyzed compounds, values of ellipticity at 208 nm and 222 nm decreased after adding every portion of **A**–**E**. These changes suggest a loss in the α-helix(%). Any shift of the peaks was not observed. The content of α-helix can be calculated using Equations (3) and (4) [28]:(3)$$\alpha -helix\left(\%\right)=\frac{-MRE_{208}-4000}{33000-4000}100\%$$
where *MRE*_208_ is the observed *MRE* value at 208 nm, 4000, and 33,000 is *MRE* value of the β-form and random coil conformation cross at 208 nm value of pure *α-helix* at 208 nm, respectively.
(4)$$MRE=\frac{ObservedCD\left(mdeg\right)}{10Cnl}$$
where *C* is the molar concentration of BSA, *n* is the number of amino acid residues, which is 583 for BSA, *l* is the path length.

All analyzed compounds caused a reduction in the α-helical contents of BSA. The results collected in Table 5 show that the biggest decreasing of α-helix(%) is observed for compound **C**. The α-helical content of BSA decreased here from 61.6% to 44.5% (change 17.1%) with increasing BSA to **C** molar ratio from 1:0 to 1:10 (Table 5, Figure 7**).** Smaller changes were observed for the interaction of BSA with **A**, **B**, **D**, and **E** compounds. Obtained values for α-helix(%) were changing from 58.7% to 52.8% (change 5.9%) for **A**, 61.8% to 52.8% (change 9%) for **B**, 59.0% to 52.3% (change 7.3%) for **D** and 61.7% to 53.9% (change 7.8%). In all cases, the changes in the molar ratio were the same as in the BSA/**C** system. Therefore, CD studies showed that all analyzed compounds bind to BSA. Obtained results are in agreement with fluorescence spectroscopy.

### 2.7. Binding Constants

For all analyzed systems, the binding constants and the binding stoichiometry were calculated. However, in the literature, there are several models to determine the binding parameters. The values obtained from different methods of calculation could significantly differ from each other [29,30,31]. Each of them also has various limitations. Two models were used to analyze the received data: double logarithm regression curve (5), modified double logarithm regression curve (6). Using the double logarithm regression curve, the binding constant K_b,_ and the number of binding stoichiometry n was determined using the following Equation (5) [23]:(5)$$log\frac{F_{0}-F}{F}=logK_{b}+nlog\left[Q\right]$$
where *F*_0_ and *F* are the steady-state fluorescence intensities at the maximum wavelength in the absence and presence of quencher, respectively, [*Q*] is the quencher concentration. The corresponding values were obtained from the slope and the intercept of the plot of log [(*F*_0_ − *F*)/*F*] versus log [*Q*]. The liner segment, corresponding to **A**–**E** /BSA molar ratio 2:1, was analyzed (Figure 6 and Table 4). The results showed that the binding constant increases in order **A**-**B**-**C**, i.e., when the phenyl ring is substituted (Figure 1). The binding constants for **D** and E are similar and lower than for **B** and **C**. The *n* value close to 1 shows one to one interaction.

The second method: modified double logarithm regression curve (6), in contrast to the previous one, takes into account the total concentration of protein present in the analyzed solution.
(6)$$log\frac{F_{0}-F}{F}=nlogK_{b}+nlog\frac{1}{\left[Q\right]-\left(F_{0}-F\right)\frac{\left[P\right]}{F_{0}}}$$
where [*P*] is BSA concentration. By the plot of log (*F*_0_ − *F*)/*F* vs. log (1/([*Q*] − (*F*_0_ − *F*)[*P*]/*F*_0_)), the binding stoichiometry n and the constant *K_b_* were obtained (Table 4, Figure 6). The results showed that the binding constants and the number of the binding site are slightly smaller than values obtained from Equation (5). It is supposed that the free concentration of the binding compound is not equal to the total concentration of the quencher.

The interaction of 14 anti-inflammatory drugs with human serum albumin was investigated by F. Mohammadnia [32]. The binding constants were found with the range 10^2^ dm^3^·mol^−1^ (acetaminophen) to 1.88 × 10^7^ dm^3^·mol^−1^ for meloxicam. So, *K_b_* values of studied compounds show that the interactions with BSA is moderate. Similar values were obtained for many compounds with biological activity. For example, series of flavonoids (binding constants in the range 1–15 × 10^4^) [33], a lipophilic derivative of thalidomide (anti-inflammatory agent) [34], a pyrimidine derivative (antibacterial reagent) [35], an indole derivative (COX inhibitor) [36] or a bis-isatin derivative (potential anti-proliferative activity) [37].

### 2.8. Thermodynamic studies

The interaction forces between a small molecule and protein include hydrogen bond, van der Waals force, electrostatic and hydrophobic interactions, etc. [38]. The forces involved in the interaction are identified by the thermodynamic analysis. The signs and magnitudes of the thermodynamic parameters identify the type of interactions [39]. The enthalpy change (Δ*H*°), the entropic change (Δ*S*°) and free energy change (Δ*G*°) were calculated from Equations (7) and (8):(7)$$logK_{b}=-\frac{\Delta H{}^{\circ}}{RT}+\frac{\Delta S{}^{\circ}}{R}$$
(8)$$\Delta G{}^{\circ}=\Delta H{}^{\circ}-T\Delta S{}^{\circ}=-RTlnK_{b}$$
where *K_b_* is the binding constant, *R* is the universal gas constant.

The calculated data for all compounds are presented in Table 6. The results showed that the binding interaction between tested compounds and BSA were spontaneous due to the negative Δ*G*° values at the studied temperature range. Furthermore, both the Δ*H*° and Δ*S*° negative values indicate that the main interaction force in the binding process was van der Waals forces and/or hydrogen bonding interaction.

### 2.9. Site Markers Studies

BSA, as well as human serum albumin (HSA), is known to possess two binding sites (a site I, site II), which are situated in subdomains IIA and IIIA, respectively [40]. To evaluate the binding site in BSA for **A**–**E**, displacement studies were carried out by using phenylbutazone (PHB) and ibuprofen (IBP) as site probes. The site I shows the binding affinity towards PHB, site II is known to bind IBP [41]. Fluorescence emission spectra of the mixed solutions of BSA and site markers following a concentration increment of **A**–**E** were recorded. Quenching rate and binding constants were analyzed using Equation (2) and (5). The results were summarized in Table 7. Results show that both K_b_ and k_q_ values of compounds **A**–**C** with BSA in the presence of IBP, considerably decrease compared to without IBP. In the case of PHB, changes were slight. It can be concluded that **A**–**C** binds to subdomain IIIA of BSA. For **D** and **E**, that binding constant in the presence of PHB and IBU considerably decline compared to without markers. It can be concluded that **D** and **E** may bind to subdomain IIA or IIIA of BSA. However, it seems that the IBP site is more preferred, especially for molecule **E**.

### 2.10. Molecular Docking-Interactions with BSA

In order to determine the preferred binding sites of the compounds **A**–**E** on BSA, the binding interactions were simulated by the molecular docking method. The simulated results were presented in Table 8. As is well known, the more negative the binding free energy ΔG°, the more stable the formed complex is. The results revealed that the binding free energy for **A**–**C** within the hydrophobic cavity in site II (subdomain IIIA) of BSA was more negative than that within the hydrophobic cavity in the site I (subdomain IIA). This indicating that site II is favorable. For **D** and **E**, the binding free energy ΔG° within site I and site II are close to each other. It indicates that both site are favorable. This result is consistent with the results observed in the site marker fluorescence studies. As presented in Table 8, the sum of van der Waals energy, hydrogen bonding energy, and desolvation free energy (ΔE_2_) is more negative than electrostatic energy (ΔE_3_). Hence, it can indicate that the main interactions between compounds **A**–**C** and BSA are van der Waals and hydrogen bonding interactions. Our thermodynamic studies also indicated that van der Waals and hydrogen bonding contributed to the interaction of the BSA-tested compound. In binding sites, II studied compounds insert into the hydrophobic cavity are surrounded by various kinds of residues (Figure 8). Hydrogen bonds with Arg208, Leu480 are formed. In the site, I pocket also hydrogen bonds are formed (Arg-217, Gly-220, Val-342). The π-sigma and other hydrophobic interactions are observed. Tryptophan residue (Trp-213) is close to tested compounds. The details are presented in Figure 8.

## 3. Materials and Methods

### 3.1. Chemistry

All the results of **C**, **H**, **N** determinations were within ±0.4% of theoretical values, carried out by Carlo Erba Elemental Analyzer model NA-1500 (Carlo Erba, Thermo Scientific, Waltham, MA, USA). ^1^H NMR spectra were determined in DMSO (**A**–**C**) or CDCl_3_ (**D** and **E**) on a Bruker 300 MHz NMR spectrometer (Bruker, Billerica, MA, USA), using TMS as an internal standard. FTIR spectra were run on Perkin-Elmer Spectrum Two, UATR FT-IR spectrometer (Perkin Elmer, Waltham, MA, USA). The samples were applied as solid.

#### 3.1.1. General Method for the Preparation of the Mannich Bases **A**–**C** (Scheme 1)

A 0.6 g (0.0031 mol) 1,6-dimethyl-3,4-pyridinedicarboximide 2 was dissolved in 40 mL of tetrahydrofurane and to this suspension 1 mL 33% formaline was added. This mixture was refluxed for 0.5 h. After this time 0.0035 mol of suitable *N*-arylpiperazines were added, again refluxed for 10 h. After one hour this mixture was cleared and other 2 more hours the products started to precipitate. The reactions were monitored by TLC. The separated solid substance was collected on a filter and washed with diethyl ether and dried (crude). The analytical samples were obtained after crystallization from distilled water (**A**) or THF (**B** and **C**). The properties of obtained compounds **A**–**C** show below (^1^H NMR plots are available in Appendix A):

**A**: 5,6-dimethyl-4-oxo-2-[(4-phenyl-1-piperazinyl)methyl]-1*H*-pyrrolo[3.4-c]pyridine-1,3(2*H*)-dione: *C*_20_*H*_22_*N*_4_*O*_3_, m.w. 366.46, m.p. 250 °C, solvent distilled water, THF, yield 93.86%; TLC R_f_ = 0.1 (ethyl acetate); R_f_ = 0.81 (ethyl acetate: methanol 1:1) ^1^H NMR δ: 2.37 (s-3*H*, *CH*_3_ at **C**-6), 2.55–2.65 (m-4*H*, -*CH*_2_-*N*-(*CH***_2_**)_2_-); 3.05–3.15 (m-4*H*, -(*CH*_2_)_2_-*N*-*C*_6_*H*_5_); 3.30 (s-3*H*, *N*-*CH*_3_); 4.39 (s-2*H*, -*N*-*CH*_2_-*N*-); 6.53 (s-1*H*, *H* arom. of pyridine); 6.70–6.75 (t-1*H*, arom *p*-*H* of benzene); 6.85–6.88 (d-2*H*, *o*-*H* of benzene); 7.13–7.18 (t-2*H*, *m*-*H* of benzene). FT-IR: *C*=*O* 1660; 1714; 1750 [cm^−1^], monosubstituted benzene 770, 700.

**B**: 5,6-dimethyl-4-oxo-2-[4-(3-trifluormethyl)phenyl-1-piperazinyl)methyl]-1*H*-pyrrolo[3.4-c]pyridine-1,3(2*H*)-dione: *C*_21_*H*_21_*N*_4_*O*_3_*F*_3_, m.w. 434.46, m.p. 267 °C, solvent THF; yield 86.29%; TLC R_f_ = 0.08 (ethyl acetate); R = 0.79 (ethyl acetate: methanol 1:1); ^1^H NMR δ: 2.37 (s-3*H*, *CH*_3_ at **C**-6), 2.55–2.65 (m-4*H*, -*CH*_2_-*N*-(*CH***_2_**)_2_-); 3.15–3.25 (m-4H, -(*CH*_2_)_2_-*N*-*C*_6_*H*_4_-*CF*_3_); 3.30 (s-3*H*, *N*-*CH*_3_); 4.40 (s-2*H*, -*N*-*CH*_2_-*N*-); 6.54 (s-1*H*, *H* arom of pyridine.), 7.10–7.37 (m-3*H*, *H* arom. of benzene). FT-IR: *C*=*O* 1660; 1714; 1757 [cm^−1^]; m-disubstituted benzene 700, 784.

**C**: 5,6-dimethyl-4-oxo-2-[4-(2-methoxy)phenyl-1-piperazinyl)methyl]-1*H*-pyrrolo[3.4-c]pyridine-1,3(2*H*)-dione: *C*_21_*H*_24_*N*_4_*O*_4_, m.w. 396.49, m.p. 234 °C, crude(insoluble on etanol, methanol; diethyl ether, CDCl_3_); yield 79.41%; TLC R_f_ = 0.06 (ethyl acetate); R_f_ = 0.75 (ethyl acetate: methanol 1:1); ^1^H NMR δ: 2.38 (s-3*H*, *CH*_3_ at **C**-6), 2.55–2.65 (m-4*H*, -*CH*_2_-*N*-(*CH***_2_**)_2_-); 2.85–2.95 (m-4*H*, -(*CH*_2_)_2_-*N*-*C*_6_*H*_4_-*OCH*_3_); 3.30 (s-3*H*, *N*-*CH*_3_); 3.71 (s-3*H*, -*O*-*CH*_3_), 4.40 (s-2*H*, -*N*-*CH*_2_-*N*-); 6.55 (s-1*H*, *H* arom. of pyridine), 6.82 -6.91 (m-3*H*, arom. of benzene). FT-IR: *C*=*O* 1660; 1710; 1750 [cm^−1^], disubstituted benzene 740.

#### 3.1.2. Method for the Preparation of the imides **D**, **E** (Scheme 1)

A 0.002 mol of 4-methoxy-6-methyl-1*H*-pyrrolo[3.4-c]pyridine-1,3(2*H*)-dion was dissolved in 40 mL of tetrahydrofurane and to this solution 0.4 mL of 33% formaline was added. This mixture was refluxed for 0.5 h. After this time 0.0022 mol of pyrrolidine was added, again refluxed for 10 h. Then, solvent was evaporated completely under reduced pressure. The residue was purified by crystallization from n-Hepxane.

**D**: 4-Methoxy–*N*-[1-(*N*-pyrrolidine)-methyl]-6-methyl-1*H*-pyrrolo[3.4-c]pyridine-1,3(2*H*)-dione *C*_14_*H*_17_*N*_3_*O*_3_, m.w. 275.35, m.p. 123 °C, solvent n-Hexane; Yeld 60%; ^1^H NMR δ: 1.691–0.76 (t-4*H*, 2*CH*_2_ pyrrolidine), 2.602–0.79 (m-7*H*, (*CH*_2_)_2_-*N* and *CH*_3_), 4,13 (s-3*H*, *OCH*_3_), 4.70 (s-2*H*, *CH*_2_), 7.19 (s-1*H*, *H* arom. of pyridine). FT-IR: *C*=*O* 1720, 1770 [cm^−1^]. (^1^H NMR plots are available in Appendix A):

A 0.02 mol of potassium was dissolved in 100 mL of anhydrous ethanol and to this solution 0.01 mol of 4-methoxy-6-methyl-1*H*-pyrrolo[3,4-c]pyridine-1,3(2*H*)-dione was added. The reaction mixture was refluxed for 15 min to the obtained suspension, next 0.012 mol of 4-(2-Chloroethyl)morpholine hydrochloride was added. The mixture was refluxed until the alkaline reaction disappeared. After filtration ethanol was evaporated to a small volume and was left to crystallize. The separated product was collected on a filter and purified by crystallization from ethanol.

**E**: 4-Methoxy–*N*-[2-(*N*-morpholine)-ethyl]-6-methyl-1*H*-[pyrrolo[3.4-c]pyridine-1,3(2*H*)-dione *C*_15_*H*_19_*N*_3_*O*_4_, m.w. 305.37, m.p. 151 °C, solvent ethanol, Yeld 50%; ^1^H NMR δ: 2.40–2.63 (m-9*H*, (*CH*_2_)_3_-*N* and *CH*_3_), 3.60–3.63 (m-4*H*, (*CH*_2_)_2_-*O* of morpholine), 3.75–3.79 (m-2*H*, *CH*_2_ α), 4.13 (s-3*H*, *OCH*_3_), 7.18 (s-1*H*, *H* arom. of pyridine). FT-IR: *C*=*O* 1710, 1760 [cm^−1^] (^1^H NMR plots are available in Appendix A).

### 3.2. Cell Line

The study was carried out using the NHDF cell line obtained from ATCC (Manasas, VA, USA). Cells were cultured at 37 °C in a humidified 5% CO_2_/95% air atmosphere incubator and passaged twice a week.

The cells were cultivated in DMEM without phenol red supplemented with 10% fetal bovine serum (FBS), 2 mM L-glutamine, 1.25 µg/mL amphotericin B and 100 µg/mL gentamicin. Prepared culture medium was stored at 4–8 °C for up to one month.

### 3.3. Tested Compounds

Tested were dissolved in DMSO to a stock concentration of 10 mM. All prepared stock solutions were stored at −20 °C for up to 6 months. For the experiment, the above-mentioned tested compounds were used in the concentration range of 10, 50, and 100 µM. Before using, all compounds were dissolved in the medium, and the final DMSO concentration did not exceed 1%.

### 3.4. MTT Assay

The MTT assay was used to measure the effect of tested compounds on the viability of NHDF cells. After incubation with the compound, the supernatant was removed, 1 mg/mL MTT solution in MEM was added, and plates were incubated for 2 h at 37 °C. The medium was then removed. Formazan crystals were dissolved in 100 µL of isopropanol for 30 min, and absorbance was measured at 570 nm using Varioskan LUX microplate reader (Thermo Scientific).

### 3.5. Cyclooxygenase Inhibition Assay

COX peroxidase activity was estimated using the ready-to-use kit (Cayman, cat. no. 701050 in triplicate for all compounds at a concentration of 100 μM. Evaluation of the peroxidase activity after 2 min incubation at RT was performed using Varioskan LUX microplate reader (Thermo Scientific, Waltham, MA, USA) at 590 nm. The results are presented as the IC_50_ values, i.e., the concentrations at which 50% inhibition of enzyme activity occurred, separately for COX-1 and COX-2. The selectivity of inhibition of cyclooxygenases was presented as ratios of IC_50_ values (COX-2/COX-1). Meloxicam was used as a reference compound with COX-2 selectivity.

### 3.6. Statistical Analysis

All results are presented as mean ± SEM (standard error of the mean) expressed as E/E_0_ ratio, where E is the culture with the addition of the tested substance, and E_0_ is the control without compound. Statistical significance was calculated compared to the control.

Due to the lack of normal distribution, the non-parametric Kruskal-Wallis test was used (with appropriate post-hoc tests). In all assays, *p* < 0.05 was used as the significance level.

### 3.7. Spectroscopic Studies

All the fluorescence measurements were carried out on a Cary Eclipse 500 spectrophotometer (Agilent, Santa Clara, CA, USA,). The interaction between synthesized compounds and bovine serum albumin (BSA) was studied in pH = 7.4 and a concentration of BSA 5.0 × 10^−6^ mol·dm^−3^. A solution of BSA was titrated by successive additions 1.0 × 10^−3^ mol·dm^−3^ solution of studied compounds, to give a final concentration 0.2 × 10^−6^–5.0 × 10^−6^ mol·dm^−3^. All experiments were measured at three temperatures: 298, 303, and 310 K. Fluorescence quenching spectra were obtained at excitation and an emission wavelength of 280 nm and 300–500 nm, respectively. The following molar ratio **A**–**E**/BSA were: 0.2–2.0 with 0.2 step, 2.0–5.0 with 1.0. Binding displacement studies were carried out in the presence of the two site markers, phenylbutazone (PHB) and ibuprofen (IBP), as sites I and II markers, respectively. Concentrations of BSA and site markers were set at 5.0 × 10^−6^ and 10.0 × 10^−6^ mol dm^−3^, respectively.

Circular dichroism spectra of all BSA solutions under simulated physiological conditions in pH 7.4 in the absence and presence of analyzed compounds were made in room temperature on Jasco J-1500 magnetic circular dichroism spectrometer (Jasco, Tokyo, Japan). 10 mm pathlength was used. CD spectra were collected at a scan rate speed of 50 nm min^−1^ with a response time of 1 s. Measurements were made in the range of 200–250 nm. All spectra were baseline corrected, and the final plot was taken from three accumulated plots. The concentrations of BSA and **A**–**E** compounds were 1.0 × 10^−6^ mol dm^−3^ and 1.0 × 10^−3^ mol·dm^−3^, respectively. Experiments were performed for BSA to each analyzed compound in molar ratios from 1:0 to 1:10.

### 3.8. Molecular Docking

The ground state geometric optimizations were calculated using density functional theory (DFT) with Becke’s three-parameter hybrid exchange function with the Lee-Yang–Parr gradient corrected correlation (B3LYP) [42,43,44] functional in combination with 6–311+G (d,p) basis set. Calculations were carried out using the Gaussian 2016 A.03 software package [45].

The high-resolution crystal structure of COX-1 and COX-2 co-crystallized with meloxicam and crystal structure of BSA were selected for docking studies (Protein Data Bank, PDB ID: 4O1Z, 4M11 [46]). The interactions with COX-1, COX-2 were performed using iGEMDOCK v.2.1 software [47]. Genetic algorithm (GA) parameters were set as 800 population size, 80 generations in 10 number of solutions. After the molecular docking, the ligand-receptor complexes were further analyzed using Discovery Studio software (http://accelrys.com/). The crystal structure of BSA with PDB ID 3V03 was obtained from Protein Data Bank (http://www.rcsb.org). All the ligands and water molecules were removed, and then hydrogen atoms were added to the protein structure. The intermolecular interactions with BSA were simulated by the molecular docking method implemented in AutoDock 4.2.6 software (http://autodock.scripps.edu/resources/references). The grid maps of dimensions 60 × 60 × 60 with a grid point spacing of 0.375 Å were calculated using AutoGrid. The centers of grid boxes were set according to the binding sites phenylbutazone (PDB ID: 2BXC) and ibuprofen (PDB ID: 2BXG) on HSA ([48]). The running times of the genetic algorithm and the evaluation times were set to 100 and 2.5million, respectively.

## 4. Conclusions

In this paper, new derivatives of pyridine-1,3(2*H*)-diones were synthesized and evaluated for their COX-I/II activity and interaction with BSA. The result of the COX-1 and COX-2 inhibitory studies revealed that all the compounds potentially inhibited COX-1 and COX-2. The selectivity index was found to be similar to meloxicam. Structural modifications did not significantly affect tested compounds activity. The docking studies were found to be in good correlation with the experimental data, and the pyrrolo-pyridine ring plays an important role in interacting with the enzyme. The experimental results showed that the fluorescence quenching of BSA by studied compounds was a result of complex formation. The values of binding constants also confirm the binding of analyzed derivatives to BSA. These results provide a valuable starting point for the design and synthesis of pyrrolo-pyridine analogs that inhibit COX as potential drugs.

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
