# Peer review of "Synthesis, Cyclooxygenases Inhibition Activities and Interactions with BSA of N-substituted 1H-pyrrolo[3,4-c]pyridine-1,3(2H)-diones Derivatives"

_molecules, 2020, doi:10.3390/molecules25122934_

Round 1

Reviewer 1 Report

The original article entitled “Synthesis, cyclooxygenases inhibition activities and interactions with BSA of N-substituted 1H-pyrrolo[3,4-c]pyridine-1,3(2H)-diones derivatives” authored by Krzyzak et al. describes the synthesis of novel compounds and their interaction with cyclooxygenase and bovine serum albumin. The manuscript is written clearly and high detail. The english language however should be checked carefully again.

The introduction gives a brief, clear and concise overview about the compound class, COX, and BSA as well as the aim of this work. Compound A was previously described in the following reference: Sladowska, Il Farmaco, 1993, vol. 48, # 1, p. 85 – 94, and should be cited and discussed accordingly. Figure 1 should also contain the lead structure which formed the basis for this study and be revised accordingly. The legend for this figure should not contain the exact name of the compounds but contain a short description instead, e.g ‘Lead and investigated compounds A-E of this study’.

The results and discussion part covers synthesis, COX inhibition and BSA binding studies combined with results of in silico and in vitro studies. Section 2.1. Chemistry contains the detailed description of experimental procedures (e.g. lines 77-82) that must be moved to the experimental section. Instead, a more general discussion should be given by the authors which covers principle reaction mechanism, yield and if applicable other issues related to the structural elucidation. Copies of 1H and 13C NMR sprectra should be provided as electronic supporting information. How N-arylpiperazines were obtained? Scheme 1 should be revised carefully, e.g. reaction arrow from 1 to 2 and conditions is missing, format of the chemical structures (same size of letters), ‘a’ and ‘b’ are not described in the legend, reaction conditions should be given in the form ‘reagents, solvent, Temp’ e.g. KOEt, EtOH, reflux. COX inhibition was found to be in the higher µM range for COX-1 and COX-2. Hence, the compounds are rather weak COX inhibitors which should be discussed, also in relation to potential applications. What other targets are known for this substance class and are they known to be more/less potent? The molecular docking studies described in detail in 2.4 are described clearly. Docking binding energies of meloxicam and comparison to A-E should be provided and discussed in relation to the observed inhibition pattern in 2.3. Fluorescence based analysis of binding to BSA are described in high detail, clearly and seem conclusive from my point, although I have to state that this is not my area of expertise. Please check the parentheses [] in line 171. Binding constants obtained for the binding to BSA should be discussed in relation to other albumin binders. Is the interaction observed comparable to other or weaker than known for other compounds?

The experimental section should be revised as follows: Chemical syntheses should be described in detail in this part of the manuscript as discussed above. Other methods are described clearly and in sufficient detail. For COX inhibition studies it was stated that measurements were performed at a concentration of 100 µM but IC50 values are given in the results and discussion part which needs determination of inhibitory potency in a centration range. Please clarify.

In the conclusion the authors state that the binding constants indicate good transport and half-life of the drug. From my point of view this conclusion should be deduced from the data as part of the discussion part with relation o literature data, otherwise the statement should be revised to the fact that compounds bind to BSA.

In conclusion, the authors presented a very detailed investigation on novel compounds with focus on their COX and BSA binding properties. While results were presented mostly clear, the implications from the obtained data should be further discussed with relation to known COX- and BSA-binders.

Author Response

Authors’ Response Reviewer #1

Dear Reviewer,

We appreciate your time and efforts in reviewing our article. Please see below our response to you comments.

  1. The manuscript is written clearly and high detail. The english language however should be checked carefully again.

The article was checked additionally using the writing assistant tool.

  1. The introduction gives a brief, clear and concise overview about the compound class, COX, and BSA as well as the aim of this work. Compound A was previously described in the following reference: Sladowska, Il Farmaco, 1993, vol. 48, # 1, p. 85 – 94, and should be cited and discussed accordingly. Figure 1 should also contain the lead structure which formed the basis for this study and be revised accordingly. The legend for this figure should not contain the exact name of the compounds but contain a short description instead, e.g ‘Lead and investigated compounds A-E of this study’.

The following text with reference has been added:

“The compound A was previously synthesized and tested for anxiolytic effects [10]. However, no activity was found, only the potential analgesic effect was suggested.”

Figure 1 has been corrected. Leading structure has been added. The figure caption has been changed.

  1. The results and discussion part covers synthesis, COX inhibition and BSA binding studies combined with results of in silico and in vitro studies. Section 2.1. Chemistry contains the detailed description of experimental procedures (e.g. lines 77-82) that must be moved to the experimental section. Instead, a more general discussion should be given by the authors which covers principle reaction mechanism, yield and if applicable other issues related to the structural elucidation. Copies of 1H and 13C NMR sprectra should be provided as electronic supporting information. How N-arylpiperazines were obtained? Scheme 1 should be revised carefully, e.g. reaction arrow from 1 to 2 and conditions is missing, format of the chemical structures (same size of letters), ‘a’ and ‘b’ are not described in the legend, reaction conditions should be given in the form ‘reagents, solvent, Temp’ e.g. KOEt, EtOH, reflux.

Thank you for this comments. Section 2.1 “Chemistry” has has been modified according to the recommendation. The description of the method and discussion about the mechanism of aminomethylation has been added. The nature of condensation using formalin has been described. Synthesis content, ordered by missing values (Rf, solvent, yield) and literature, has been moved to Section 3 Materials and methods (line 349, 350 et more). In section 2.1. discussion on the properties of starting compounds 2 and 3 and model imide 1 is provided. Copies (PDF) NMR spectra (supplementary) are attached. N-arylpiperazine mentioned in the text [CAS: 92-54-6; 35-386-24-4; 15532-75-9] and 4- (2-chloroethyl) morpholine hydrochloridum [CAS:3647-69-6] were purchased from Sigma-Aldrich.

Scheme 1, concerning the synthesis, has been ordered, supplemented with a description of the methods in a unified form and order.

The following text with reference has been added:

Synthetic scheme of the studied compounds, as shown in Scheme 1. The compounds 2,3 and A were obtained according to the literature data [10–13]. Imide 2 is the starting product for derivatives A, B and C and many other substances described earlier [10,14,15]. Compounds 2 and 3 differ slightly in structure, imide 3 contains a methoxy substituent attached to the carbon atom in a position adjacent to the nitrogen atom of the pyridine ring, while derivative 2 is its N-methylanalogue. To compare the biological properties of both basic systems, it was decided to synthesize analogues of the previously described derivatives containing the same amino residues: 1-phenylpiperazine (A), 1-(2-methoxyphenyl)piperazine (B) and 1-(3-trifluoromethyphenyl)piperazine (C) [16]. The presence of an acidic proton atom at the imide nitrogen atom in position 2 makes it possible to carry out the aminomethylation reaction according to the mechanism described by Mannich [13,17]. The reaction was carried out using aqueous solution of formaldehyde (HCHO) and N-arylpiperazines (commercial products Sigma-Aldrich) at the reflux temperature of tetrahydrofuran (THF) for several hours (method I, scheme 1). The course of condensation was monitored by TLC. During the reaction of (A,B) or after the evaporation of the solvent (C) final products were obtained. Poorsolubility in organic solvents (chloroform, ethanol, methanol, ethyl acetate) and their mixtures prevented the purification of the products by column chromatography, therefore the final compounds were tested in the form of crude products. NMR spectral analysis was performed in dimethyl sulfoxide (DMSO). All reactions proceeded with a very good yield of 79.4-93.8%.

Imid 3, previously described [11,13] was the starting product for the synthesis of several dozen derivatives with confirmed biological activity (Śladowska et al.) [13,16,18,19].

The main line of analogues was based on the structure of 4-alkoxy-1H-pyrrolo(3,4-c) pyridine-1,3(2H) -dione. Optimal biological properties in the behavioral tests showed imide1: 4-alkoxy-N- [3- (N-phenyl-4-piperazinyl) -2-hydroxy] propyl-1H-pyrrolo (3,4-c) pyridine-1, 3(2H)-dione. One of the structure modifications is the assumption of shortening to two (E) or one (D) methylene groups of the alkyl linker between the 2-methoxy-3,4-pyridinedicarbox imide moiety and the arylamine residue. In order to obtain D and E derivatives, commercial preparations of 4-(2-chloroethyl)-morpholine and pyrrolidine were used, condensing the potassium salt of imide 3, under the influence of potassium ethoxylate in anhydrous ethanol (method II, scheme 1 ; E), or in the Mannich reaction - THF - solvent and formalin ( method III, scheme 1; D). The reaction yield was 50% (E) and 60% (D).”

  1. COX inhibition was found to be in the higher µM range for COX-1 and COX-2. Hence, the compounds are rather weak COX inhibitors which should be discussed, also in relation to potential applications. What other targets are known for this substance class and are they known to be more/less potent?

We would not say that "COX inhibition was found to be in the higher µM range". In our opinion, for all compounds, the level of COX-1 and COX-2 inhibition was comparable to meloxicam. We have shown that the compounds are COX inhibitors not worse than meloxicam. Potential applications of the studied substance class were mentioned in the “Introduction” section. We are looking for substances from which we do not necessarily expect better activity. It would be sufficient to reduce the side effects (e.g. gastrointestinal toxicity) compared to known drugs.

  1. The molecular docking studies described in detail in 2.4 are described clearly. Docking binding energies of meloxicam and comparison to A-E should be provided and discussed in relation to the observed inhibition pattern in 2.3.

Binding energies of meloxicam for interactions with COX-1 and COX-2 have been added to Table 2.

“The results indicated that compound D showed the lowest total binding energy Etotal with respect to COX-1. The highest value was observed for compound E. The docking scores showed a good correlation with IC50 values (Table 1). For interactions with COX-2 Etotal was more negative. The lowest energy was observed for A and E and the highest for C. This result was also in good correlation with inhibitory activity. The compounds with the lowest binding energy showed the most promising biological activity in cyclooxygenase inhibition assay. The molecular docking results also indicated that binding energy of tested compounds were more negative than for meloxicam.”

  1. Fluorescence based analysis of binding to BSA are described in high detail, clearly and seem conclusive from my point, although I have to state that this is not my area of expertise. […]. Binding constants obtained for the binding to BSA should be discussed in relation to other albumin binders. Is the interaction observed comparable to other or weaker than known for other compounds?

The following text with reference has been added:

„The interaction of 14 anti-inflammatory drugs with human serum albumin was investigated by F. Mohammadnia [32]. The binding constants were found with the range 102 dm3mol-1 (acetaminophen) to 1.88x107 dm3mol-1 for meloxicam. So, Kb values of studied compounds show that the interactions with BSA is moderate. Similar values were obtained for many compounds with biological activity. For example, series of flavonoids (binding constants in the range 1–15x104) [33], a lipophilic derivative of thalidomide (anti-inflammatory agent) [34], a pyrimidine derivative (antibacterial reagent) [35], an indole derivative (COX inhibitor) [36] or a bis-isatin derivative (potential anti-proliferative activity) [37].”

  1. [...] For COX inhibition studies it was stated that measurements were performed at a concentration of 100 µM but IC50 values are given in the results and discussion part which needs determination of inhibitory potency in a centration range. Please clarify.

The experiment was carried out in accordance with the manufacturer's instruction: https://www.caymanchem.com/pdfs/701050.pdf (page 11). Based on the results obtained for a concentration of 100 µM, the concentration at which 50%  inhibition of enzyme activity occured was determined. We know that it would be better to do more measurements for different concentrations, but unfortunately we were forced to simplify this for financial reasons.

  1. In the conclusion the authors state that the binding constants indicate good transport and half-life of the drug. From my point of view this conclusion should be deduced from the data as part of the discussion part with relation o literature data, otherwise the statement should be revised to the fact that compounds bind to BSA.

It has been changed in conclusion: “The values of binding constants also confirm the binding of analyzed derivatives to BSA”

Reviewer 2 Report

This paper describes chemical synthesis, COX inhibition and BSA binding of five 1H-pyrrolo[3,4-c]pyridine-1,3(2H)-dione derivatives. This contains several novel information and can be accepted for publication after a minor revision considering the following comments.

Table 1. COX-1 and COX-2 inhibition of Meloxicam is 83.7 microM and 59.2 microM respectively and COX selectivity is 0.71. However, Boehringer-Ingelheim Japan officially adopts the data of Ogino, K. et al., Pharmacology 55 (1), 44-53, 1997. According to Ogino, COX-1 and COX-2 inhibition is 142 microM and 11.8 microM respectively and hence COX selectivity is 0.08 (much better than the Author’s selectivity). Authors must refer to the Ogino’s data here. As the Ogino’s experimental procedure is not the same as the Author’s procedure, the reviewer imagines that the data of A, B, C, D, E in Table 1 might be much better by employing Ogino procedure.

There are two main serum proteins, serum albumin (SA) and alpha1-acid glycoprotein (AAG). It is well-established that SA binds acidic drugs and AAG binds basic drugs. Structurally, compounds A, B, C, D, E are basic (at least not acidic) and appear typical AAG-binding structures. As AAG is induced by inflammation, the Authors must consider AAG.

Author Response

Authors’ Response to Reviewer #2

Dear Reviewer,

We appreciate your time and efforts in reviewing our article. Please see below our response to your comments.

  1. Table 1. COX-1 and COX-2 inhibition of Meloxicam is 83.7 microM and 59.2 microM respectively and COX selectivity is 0.71. However, Boehringer-Ingelheim Japan officially adopts the data of Ogino, K. et al., Pharmacology 55 (1), 44-53, 1997. According to Ogino, COX-1 and COX-2 inhibition is 142 microM and 11.8 microM respectively and hence COX selectivity is 0.08 (much better than the Author’s selectivity). Authors must refer to the Ogino’s data here. As the Ogino’s experimental procedure is not the same as the Author’s procedure, the reviewer imagines that the data of A, B, C, D, E in Table 1 might be much better by employing Ogino procedure.

The research carried out by Ogino in 1996 was an in vivo study that much better reflects the effects of a substance in the body. We only used the enzymatic assay. As the Reviewer suggests, it is very likely that according to the Ogino procedure the results would be better.

Of course, in the lines 112-116 we've added a reference to Ogino’s results.

The following text with reference has been added:

“The IC50 values obtained for meloxicam are significantly higher than those published by Ogino et al [20]. However, it was an in vivo study in rats, and we performed only the enzymatic assay. IC50 values for tested compounds are similar to those of meloxicam. Therefore in the next stage of research, it is planned to conduct in vivo tests to check the anti-inflammatory properties and gastrointestinal toxicity of the most promising compound E”.

  1. There are two main serum proteins, serum albumin (SA) and alpha1-acid glycoprotein (AAG). It is well-established that SA binds acidic drugs and AAG binds basic drugs. Structurally, compounds A, B, C, D, E are basic (at least not acidic) and appear typical AAG-binding structures. As AAG is induced by inflammation, the Authors must consider AAG.

Thank you very much for this comment. Indeed the research carried out with alpha1-acid glycoprotein would be very valuable. But the analysis of the interaction between this protein would be a very large another part of the experiment, forming in our opinion rather separate second manuscript than next chapter in this paper. We believe that the only right approach would be to perform all spectroscopic measurements and molecular modeling study, like in case of BSA. Furthermore, during the 5 days we received for the review, it is not possible to perform so many measurements, their interpretation and description. But we will definitely use this suggestion in our future research.

Reviewer 3 Report

Generally, I have not serious comments, but:

In the Abstract: English should be improved. In particular, „drugs produce activity” is not a good expression. Besides, the same sentence we can found in other publications:„anti-inflammatory drugs produce their therapeutic activities through inhibition of cyclo-oxygenase”-in The Pharmaceutical and Chemical Journal, 2017, 4(1):16-24, The Pharmaceutical and Chemical Journal (authors: Omodamiro & Jimoh). Here, authors did not include any reference (what is understandable-this is abstract)-but it could be considered as plagiarism. Authors should eliminate such unambiguous situations in all the text!!!

Furthermore, other examples of discontinuities are as follow:

Line 21: „ Interaction with BSA was studied fluorescence spectroscopy and circular dichroism measurement” should be replaced by …..” studied by (or using) fluorescence spectroscopy…”

Keywords: semicolons should be present (not comma).

Line 65: this is the first place with an explanation of BSA-it could be earlier.

Abbreviations are absent.

Figure 8 is absent, maybe Figure 9 should be Figure 8?

The quality of Fig. 1and Fig. 6  could be better.

Line 171: exposed to the solvent [].  (reference no. 14 is missing?)

Line 177, 226, 229, 265: equation (1) , (3) & (4), & (6)–particularly fraction is defective

Line 501: the year 2006 should be bold

Line 512: a year – bold and without the semicolon

Similar situation in Line 536: correct, please

Please, check carefully all references and text.

Author Response

Authors’ Response to Reviewer #

Dear Reviewer,

We appreciate your time and efforts in reviewing our article. Please see below our response to your comments.

"The article ha been checked and corected accordind to the comments. The abbreviations section has been add."